# Structure Based Dataset on SAT Solving with Graph Neural Networks

**Yi Fu**
The University of New South Wales
yi.fu.1@student.unsw.edu.au

**Anthony Tompkins**
The University of New South Wales
anthony.tompkins@unsw.edu.au

**Yang Song**
The University of New South Wales
yang.song1@unsw.edu.au

**Maurice Pagnucco**
The University of New South Wales
morri@cse.unsw.edu.au

## Abstract

Satisfiability (SAT) solvers based on techniques such as conflict driven clause learning (CDCL) have produced excellent performance on both synthetic and real world industrial problems. While these CDCL solvers only operate on a per-problem basis, graph neural network (GNN) based solvers bring new benefits to the field by allowing practitioners to exploit knowledge gained from previously solved problems to expedite solving of new SAT problems. However, one specific area that is often studied in the context of CDCL solvers, but largely overlooked in GNN solvers, is the relationship between graph theoretic measure of *structure* in SAT problems and the *generalisation* ability of GNN solvers. To bridge the gap between structural graph properties (e.g., modularity) and the generalisability (or lack thereof) of GNN based SAT solvers, we present **StructureSAT**: a curated dataset, along with code to further generate novel examples, containing a diverse set of SAT problems from well known problem domains. Furthermore, we also utilise a novel splitting method that focuses on deconstructing the families into more detailed hierarchies based on their structural properties. With the new dataset, we aim to help explain problematic generalisation in existing GNN SAT solvers, and demonstrate an alternative approach to expedite GNN training efficiency by exploiting knowledge of structural graph properties. We conclude with multiple future directions that can help researchers in GNN based SAT solving develop more effective and generalisable SAT solvers.

## 1 Introduction

The satisfiability (SAT) [10] problem is a hallmark of computer science research with remarkable real-world utility, especially in solving combinatorial optimisation problems. SAT has formed the basis of the study of computational complexity especially around the complexity class of NP problems. Moreover, as a theoretical tool, it has facilitated research into the nature of computation and solving difficult computational problems. On the practical side, SAT has been applied to many interesting real world problems such as logistics planning [36], product configuration [56], and software verification [34], retaining its high relevance today.

Advances over the last decade in SAT solving have appeared to converge on conflict driven clause learning (CDCL) methods [46] for the best general problem solving performance. While it is widely believed that CDCL solvers are performant towards various aspects of problems [2], such solvers almost exclusively operate on a per-problem basis. That is to say they do not explicitly reuse knowledge from different problems. Alternatively, graph neural networks (GNNs) have emerged as a

38th Conference on Neural Information Processing Systems (NeurIPS 2024).

complementary approach to representing and solving SAT problems by incorporating the benefits of deep learning [26]. Using an optimisation based approach, as opposed to a pure search like algorithm in CDCL, GNN-based solvers have the potential to adapt useful information from training problems to accelerate solving unseen novel problems.

However, SAT problems largely reside in NP-hard problems. In reality, for machine learning methods such as GNNs, prior works have demonstrated provably negative results on challenges that are NP-hard [62]. Although deep neural networks have the ability to ingest large datasets [39, 55], there lacks a dataset that facilitates sufficiently large and unbiased training for SAT problems, and existing SAT datasets with higher difficulties generally have a limited number of problems. Indeed, currently the largest benchmarks SATLIB [32] and SATCOMP [33] contain less than 10k industrial problems. While we can generate synthetic datasets to train GNNs [53], the resulting models would struggle to generalise to more diverse and challenging, or real-world problems [44], limiting the application of GNN-based solvers. Moreover, we believe generalisation in the context of SAT problems has currently been greatly underrated and oversimplified. More specifically, SAT problems in prior work are typically generated in a random manner without delving into what makes instances difficult or useful for training, potentially limiting the models from generalising to more challenging datasets, especially to industrial instances. For example, the largest SAT dataset on GNN – G4SATBench [44], which is constructed with 7 generators, only considers generalisation on the numbers of variables as a measure of performance vs. complexity. However, the impact of the *quality* of a dataset is greatly overlooked and remains a less explored area. In particular, problem difficulty can be strongly influenced by the intrinsic *graph structure* of each problems, as shown experimentally on traditional solver [2]. No prior work has shown if the same structural properties studied in SAT so far could influence GNN's performance, especially their ability to generalise.

To bridge the gap between structural measures and generalisability of GNNs on SAT, we propose StructureSAT, a large-scale dataset containing diverse problem domains and structural measures. In this work, we are not interested in the set of all possible SAT problems, but rather existing, well studied, SAT problem domains which we aim to analyse through the lens of graph theoretic structure. Thus, we focus on small, easy-to-generate synthetic training datasets for developing GNN models that can better generalise. StructureSAT contains 11 SAT domains from 4 high level categories: random, crafted, pseudo-industrial and industrial, within which we study 9 structural properties that have proven to be influential to traditional SAT solvers, including both conjunctive normal form (CNF) based and graph based properties [2]. We carefully deconstruct each problem domains based on their structural properties, and split each domain into low-value property subsets and high-value property subsets for each of the properties. Details of the dataset can be found in Appendix B. Moreover, to capture the relationship between the SAT problem structures and generalisation ability of GNN solvers, we conduct thorough experiments using 3 GNN baselines and benchmark their generalisation performance on a diverse set of in-domain and out-domain problems. Our results highlight the existing challenge and potential future directions on improving the generalisation performance of current GNN-based solvers. We bring light to training set to test set generalisation in GNN-based SAT solvers, and graph theoretic notions of structure, which have been well studied in the CDCL SAT literature but only lightly touched on for GNN-based SAT solving.

## 2 StructureSAT

StructureSAT consists of multiple problem domains, for each of which we control the corresponding attribute values. In this section, we briefly explain SAT problems and present the structure properties measured in the dataset. Then, we introduce the types of data and generators used for raw data generation. Lastly, we detail our structure-based splitting method. Detailed explanations of our dataset are presented in Appendix B.

### 2.1 SAT preliminaries

In propositional logic, the SAT problem is the problem of finding an assignment of truth values (true or false) to propositional variables that make a Boolean formula satisfiable (i.e., true). Boolean formulae are typically expressed in conjunctive normal form (CNF). We denote the set of propositional variables by $V$. A literal $l$ is either a variable $v \in V$ or its negation $\neg v$ (or $\overline{v}$). A clause $c$ is a disjunction of literals $(l_1 \vee l_2 \vee ..l_n)$, where $\vee$ denotes propositional logic "or". A formula $f$ is a conjunction of

clauses $(c_1 \wedge c_2 \wedge ...c_n)$, where $\wedge$ denotes propositional logic "and". Generally speaking classical SAT solvers can be divided into *complete solvers* and *incomplete solvers*. A complete solver is able to prove unsatisfiability or find a satisfying assignment if they exist for a problem. Most complete solvers are based on the Davis–Putnam–Logemann–Loveland (DPLL) algorithm [18, 17], a backtracking search algorithm. Two popular variations of DPLL solver are the Conflict-Driven Clause Learning (CDCL) solvers [46], which learn and add conflict clauses to the original formula, and look-ahead solvers [31], which do lookahead for a selected decision variable. In contrast, an incomplete solver cannot prove unsatisfiability such as the Moser-Tardos (MT) solver [12] and WalkSAT solver [51] that are derived from the stochastic local search (SLS) algorithm, which repeatedly selects variable and changes its value until an assignment or a limit is reached.

SAT formulas have been expressed using various graphs focusing on different structural properties [2]. Traditional SAT community usually encodes SAT formulas as undirected weighted graph including *variable incidence graph* (VIG) and *clause-variable incidence graph* (CVIG) [7]. VIG is a graph with literals as nodes and two literals are connected with edges if and only if they occur in the same clauses. CVIG is a bipartite graph having the set of variables and clauses as vertices. They are connected if a variable occurs inside a clause. Another commonly used set of graphs are unweighted graphs including VIG, *variable clause graphs* (VCG), *literal incidence graphs* (LIG), and *literal clause graphs* (LCG). LIG extends VIG through extra edges between literals and their negations. VCG and LCG are bipartite graphs similar to CVIG, but without weights. LCG also has an extra edge between literals and their complements. Among all the graphs, LCG has been used most in GNN based SAT solving as others either lose important information of clause (LIG and VIG), or information of polarity (VCG). As prior research [44] has shown that the particular bipartite graph construction (LCG or VCG) does not have much effect on model accuracy while non-bipartite graphs (LIG or VIG) , we will be focusing on LCG only.

## 2.2 SAT structure properties

In this work we classify structural properties of SAT problems into two classes based on the embedding methods: CNF based properties and graph based properties [1]. These properties are used as splitting method for training and testing in the dataset.

Given a SAT problem $P$, the **CNF based properties** are related to the CNF encoding of $P$. The most popular ones are **Backbone** [37] and **Phase transition** [14]. **Graph based properties** are structural properties from embeddings of a *LCG* graph $G$. Given a problem with set of variables $V$ and set of clauses $C$, the LCG graph $G$ with vertex $X$ and weight $w$ is defined as $G = (X, x | x \in C \wedge V, w)$ and the weight is either $1/|X|$ if variable is in clause, $1/|V|$ for weights between variables and their negations, or 0 if nodes are not connected. Each node $x$ in graph has degree $deg_x$. Included graph properties in StructureSAT are **Self-similary** [3], **Scale-free** [6], **Treewidth** [47], **Centrality** [22], **Community structure** [5], **Small-world** [58] and **Entropy** [66].

## 2.3 StructureSAT composition

We use various existing SAT generators and datasets to produce original SAT formulas in 11 domains. These include **Random-3-SAT** generated using CNFgen [40], **SR(*n*)** [53], **Combinatorial problems** (5 domains) from CNFgen[40], **Community Attachment** (CA) from [24], **Popularity-Similarity** (PS) [25], industrial problems **SATCOMP** from SATCOMP2007 and psudo-industrial problems from Graph Generative Models **G2SAT** [63].

## 2.4 Structure-aware splitting and generalisation selection

We also propose a novel splitting method for StructureSAT and focus on 3 types of generalisation tasks. After generating the raw datasets, instead of randomly splitting them to train/valid/test sets, we divide each raw datasets based on specific structural values. Specifically, for each domain, we select the average value $Z$ from every calculated graph-based properties, as shown in Table 3, and split each domain into a low-value subset and a high-value subset, producing a total of 16 subsets (8 structures $*$ 2 ranges) per domain. Each subsets contains 80k training data and 10k validation data.[1]

---

[1] Our dataset and codebase are available here.

For different testing sets, we focus on 3 aspects of test problems: in-domain larger problems, in-domain problems with different properties, and out-of-distribution problems. Specifically, for in-domain larger problems, we follow the random generation process and generate larger problem sets for all the synthetic generators. For example, for training with SR problems, the training and validation sets contain problems with 10-40 number of variables, while the testing sets contain problems with 40-100 and 100-200 number of variables, each with 10k pairs.

To test generalisation on in-domain problems with different structures, we are interested in random 3-SAT with different $c/v$ ratios and different backbone values on all domains. For analysing the CNF based property - $c/v$ ratio on random 3-SAT, the training and validation sets are problems with 10-40 number of variables, and a $c/v$ ratio of roughly 4.7. Our testing set contain random 3-SAT problems with same number of variables, but different number of clauses. Specifically the test set has a $c/v$ ratio of 3.5, 4, 5.5 and 6, with each ratio having 10k pairs of problems.

For out-of-distribution problems, we randomly generate 10k problem pairs from each synthetic generators without property split, while also including small-size industrial and application problems from SATCOMP and generated G2SAT problems.

We also empirically analyze the effect of augmentation to generalization, since this might cause the destroying of structure during the addition of learned clauses [5]. Prior work [44] has shown that training on problems with adding learned clauses from CDCL solvers could lead to better accuracy on augmented SAT domains than training on raw problems. Thus, we gather learned clauses from produced DRAT-trim proofs [60] after running problems with the CadiCal solver [21], then augment our dataset with the collected learned clauses.

## 3 Experiment

In this section, we experimentally evaluate the out-of-distribution generalisation ability of GNN-based SAT solvers with StructureSAT. Specifically, we investigate the following questions: Q1: How does the structure influence generalisation to larger size problems; Q2: Could GNNs generalise to the same domain but for problems with different structure; Q3: How much does structure influence out of distribution generalisation; Q4: Does training on augmented problems influence out of distribution generalisation? Due to the limit in space, we will present the experiment on Q3 and Q4 only in the following section. All other experiments can be found in Appendix F.

### 3.1 Evaluation setup

With the newly developed dataset, with details we conduct several experiments on the task of predicting satisfiablity of SAT problems, which is considered as supervised binary graph-classification task. We consider 3 baseline GNN models NeuroSAT [53], GCN [38] and GIN [61] in our experiments and follow the implementation of G4SATBench [44]. We refer the reader to Appendix E for more implementation details. As prior work has shown that different GNN models perform similarly in general [44], we want to emphasise that the main focus of this paper is to find the general trend on generalisation performance of GNN-based SAT solvers with regard to various structural measures, which can guide future development of robust GNN-based solvers applicable to challenging problems.

### 3.2 Generalisation to other domains

**Out-of-distribution generalisation without augmentation**    To answer Q3, we train NeuroSAT on the small and big split of SR(10-40), and test on different domains without splits. As presented in Table 1, the results differ vastly across the structures and domains. Taking $D_f$ as an example, both $k$-clique and PS have similar $D_f$ values. However, testing results on PS are significantly better than on $k$-clique. Furthermore, models trained on larger $D_f$ group seems to do better on PS, while PS has a larger $D_f$ value than SR(10-40) on average. This suggests that training on problems with similar metric as testing problems could possibly increase GNN performance. However, it contradicts the fact that smaller $\alpha_v$ reaches better performance on CA, as CA dataset has bigger $\alpha_v$ value than SR(n). One possible reasons for this contradiction would be that the performance of GNN is not determined by one single metric property, but rather a combination of properties.

Table 1: Accuracy of NeuroSAT trained on SR(10-40) with different structural splits.

| Metric | Split | Testing Domains | | | | | | | | | | |
|--------|-------|------|------|------|------|------|------|------|------|------|------|------|
| | | SR | R3 | KCL | KD | KV | KCO | AM | CA | PS | G2 | IN |
| $D_f$ | small | 94.49 | 91.75 | **55.86** | **60.36** | **64.31** | **44.66** | 45.66 | **82.33** | 91.54 | **95.83** | 58.33 |
| | big | **95.43** | **93.36** | 50.00 | 50.22 | 50.00 | 44.47 | **49.31** | 71.2 | **95.98** | 91.67 | **75.00** |
| $\alpha_v$ | small | **94.40** | **91.72** | 48.76 | **66.03** | **74.89** | 44.70 | 46.98 | **90.28** | **94.44** | **95.83** | 50.00 |
| | big | 93.05 | 91.40 | **51.60** | 53.21 | 53.64 | **56.14** | **58.37** | 70.51 | 91.11 | 83.33 | 50.00 |
| $\alpha_c$ | small | **95.41** | **92.99** | **51.85** | **59.27** | **68.10** | **55.97** | 43.85 | 92.36 | 96.04 | 87.50 | 58.33 |
| | big | 95.31 | 92.82 | 48.83 | 56.14 | 54.97 | 51.75 | **45.79** | **93.31** | **96.06** | **91.67** | **75.00** |
| $T_w$ | small | 95.04 | 91.79 | **51.15** | 54.49 | **57.60** | **52.37** | 44.42 | **74.56** | 94.05 | 91.67 | 66.67 |
| | big | **95.33** | **92.79** | 44.46 | **65.40** | 55.61 | 51.85 | **47.41** | 50.01 | **94.14** | 91.67 | 66.67 |
| $Q$ | small | 93.76 | 91.87 | **56.30** | **71.76** | 51.9 | 55.89 | **49.75** | 89.08 | **95.75** | 79.00 | 50.00 |
| | big | **94.81** | **92.74** | 51.61 | 61.06 | **60.80** | **61.64** | 46.57 | **89.99** | 93.00 | **83.33** | **58.33** |
| $H$ | small | 94.00 | 91.72 | **52.44** | 66.79 | **60.48** | 50.39 | 36.97 | **89.88** | 91.41 | **91.67** | 50.00 |
| | big | **95.77** | **92.27** | 48.62 | **68.4** | 58.27 | **55.97** | **41.93** | 77.89 | **92.18** | 87.50 | **58.33** |

Table 2: Augmented experiments. Top: NeuroSAT trained on augmented datasets. Testing dataset is split to augmented and raw. Bottom: Average graph based properties before and after augmenting.

| Domain | Split | SR | R3 | KCL | KV | KD | KCO | CA | PS | G2 | IN |
|--------|-------|------|------|------|------|------|------|------|------|------|------|
| SR | augment | **99.99** | **99.99** | 50.00 | **58.90** | **64.00** | **94.40** | **70.40** | **99.58** | 4.17 | **66.67** |
| | raw | 50.00 | 50.00 | 50.00 | 55.96 | 54.00 | 57.40 | 50.00 | 51.05 | 4.17 | 41.67 |

| Domain | Split | $D_f$ | $\alpha_c$ | $\alpha_v$ | $Q$ | $T_w$ | $B_e$ | $H$ |
|--------|-------|-------|------------|------------|------|-------|-------|------|
| SR | augment | 2.46 | 4.24 | 9.78 | 0.39 | 39.16 | 0.013 | 5.03 |
| | raw | 3.39 | 4.25 | 10.03 | 0.38 | 39.12 | 0.012 | 5.01 |

**Augmented problem generalisation.** To get insights from Q4, we augment SR(10-40) with learned clauses from Cadical solver. Then, we test the results on different domains. Both training and testing data here are not splitted with structural properties. Although models trained on augmented datasets reach high accuracy in other augmented domains, with only a few learned clauses added (on average the ratio between learned clause and original clause is roughly 15.9% in training data), they could not solve any raw data well in any domains. While most structural parameters remain the same before and after augmentation, the relatively large change in $D_f$ could be one reason for this result.

## 4 Discussion and Conclusion

This paper presents StructureSAT, a structure-based SAT dataset with various domains and property measures. We aim to help analyze GNN based generalisation in SAT solving by adapting a variety of structural properties. Through our extensive cross-domain experiments using StructureSAT, we produced useful insights in GNN generalisation with empirical analysis. As suggested from our experiments, certain properties are more influential on generalisability than other properties. Furthermore, the diverse GNN generalisation abilities on different domains could be the result of a combination of properties. We hope StructureSAT brings interest into the role of structural properties for future research into GNN based SAT solving.

LLM: With the emergent of LLM in logic and reasoning, a large dataset with various domains are needed for training and testing in the field. StructureSAT could help LLM on the task of solving SAT problems by providing data with large quantity covering wide domains, and the task of structure prediction on SAT, which can be further combined with traditional SAT solving on solver selection.

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

# A    Related work

**SAT dataset.** SAT problems can be classified into different categories, mainly as random, crafted, and industrial [1]. Research studies related to SAT either generate synthetic instances using generators or use existing datasets. Most generators focus on similar type of problems like random $k$-SAT and combinatorial problem generators like CNFgen [40]. The largest established datasets are SATLIB [32] and the yearly SAT Competitions (SATCOMP) [33]. To overcome the limitation on number of instances in existing industrial dataset, new generators have been proposed to generate pseudo-industrial SAT instances, including hand-crafted generators Community Attachment (CA) [24] and Popularity-Similarity (PS) [25], or graph-generative models generators [63, 43, 42, 15, 23]. For GNN based SAT solving, [53] proposed random generator SR($n$) and [11] generated uniform-random 3-SAT instances as datasets. G4SATBench [44] produced a dataset including 7 types of synthetic generators with varying variable sizes. Following G4SATBench, our dataset extend to 10 types of synthetic generators with better quality and hardness measures from structural values. We also include industrial dataset for more generalisation test and analysis on real-world application.

**SAT structural properties.** As traditional SAT solvers perform differently over random, crafted, and industrial instances, it is believed that different SAT domains have distinct underlying properties [2]. Results in this filed have been wildy used in improving traditional solvers [8] and portfolio based solvers [57], classifying benchmark [4], generating instances [25], and defining problem hardness [49]. Thus, many attempts have been made to define, prove, and analyze the structural measures within SAT, including problem-based properties and solver-based properties. Problem-based properties are further divided to CNF-based, including phase-transition [14], backdoor [37] and backbones [37], and graph-based, including scale-free [6], self-similar [3], centrality [35], treewidth [47], entropy [66], small-worlds [58] and community structure [5]. Solver-based properties are directly related to SAT solver such as: mergeability and resolvability [67]. These measures are either proved mathematically, or analyzed through solver-related parameters such as solving time. However, all the works are experimented with traditional SAT solver. Furthermore, there is little work on multiple properties [41, 67] while most works only focus on single measure and benchmark. In this work, we select several related measures from [1], and calculate their values on each of our selected domains. We also propose new splitting method of our dataset to high-value subset and low-value subset of each structure, while analyzing them against cross-domain generalisation result on GNN, concluding the important and hard features for GNN to capture.

**GNN for SAT solving.** GNN has been applied to SAT solving mainly as problem solvers, including standalone solvers and hybrid solvers. Standalone solvers are networks trained to classify problems as satisfiable or unsatisfiable themselves, which mainly encodes the formula as LCG graphs [53, 64, 13, 54, 11, 30, 19, 50]. On the other hand, hybrid solvers treat GNN as a guidance to traditional solvers by replacing their heuristics with network predictions. These solvers focuses on predicting specific tasks such as UNSAT core [52], glue clauses [29], backbone variables [59], for CDCL solvers and initial assignment [65, 45] for SLS solvers. Although hybrid solvers generally achieve better results than standalone solvers, they acts as modifications of traditional solvers instead of discussing solvability of GNNs, which is out of scope of our work. Furthermore, for most GNN-based solvers, only the ability to generalise to larger in-distribution problems is discussed, which ignores the underlying structure of each domain. In this work we train several GNNs as standalone solvers from [44], and combine the structure properties of SAT with out-of-distribution generalisation on GNN.

# B    Dataset description

StructureSAT consists of multiple problem domains, for each of which we control the corresponding attribute values. In this section, we start by presenting the structure properties measured in the dataset. Next, we introduce the types of data and generators used for raw data generation. Last but not least, we detail the construction process of our dataset, our structure-based splitting method, and the generalisation tasks we focus on.

## B.1    SAT structure properties

In this work we classify structural properties of SAT problems into two classes based on the embedding methods: CNF based properties and graph based properties [1]. These properties are used as splitting

method for training and testing in the dataset. While there are usually 4 types of graphs to represent SAT problems Fig. 1, we use the LCG graph in this work due to having the most amount of information among all 4 types.

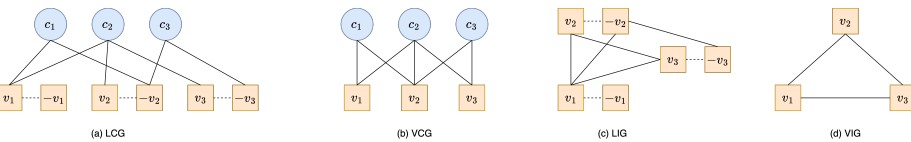

Figure 1: Different Graph representation of SAT problems.

**CNF based properties.** Given a problem $P$ with set of variables $V$, the CNF based properties are related to the CNF encoding of a problem.

- **Backbone** [37] refers to the set of literals of a problem which are true in all satisfying assignments. The value of each literals in the set is fixed in all assignments of the problem. StructureSAT calculates the size of the backbones($B_b$) of satisfiable instances only.

- **Phase transition** [14] is a phenomenon measured by clause to variable ratio ($n_c/n_v$), where $n_c$ and $n_v$ represents number of clauses and variables respectively. It is evident that an easy-hard-easy pattern occurs for random $k$-SAT where transition for random 3-SAT is at $c = 4.258 * n + 58.26 * n^{-1/2}$ [16]. Although the phenomenon has been observed in both random $k$-SAT and peudo-industrial instances, we will only be focusing it on random 3-SAT in this dataset.

**Graph based properties** are structural properties from embeddings of a *LCG* graph $G$. The definition is extended from the CVIG definition from [7]. Given a problem with set of variables $V$ and set of clauses $C$, the LCG graph $G$ with vertex $X$ and weight $w$ is defined as $G = (X, x | x \in C \land V, w)$ and the weight is either $1/|X|$ if variable is in clause, $1/|V|$ for weights between variables and their negations, or 0 if nodes are not connected. Each node $x$ in graph has degree $deg_x$.

- **Self-similary** [3] is measured by fractal dimension($D_f$). $G$ is self-similar if the minimum number of boxes of size $s$ required to cover $G$ decreases polynomially for some $D_f$.

- **Scale-free** [6] can be measured by frequency of variables ($\alpha_v$) or clauses size ($\alpha_c$). A graph is scale-free if the arity of nodes is characterized by a random variable N that follows a power-law distribution.

- **Treewidth** ($T_w$) [47] measures the tree-likeness of graphs. It is the minimum width over all possible tree decomposition of G. In this work we calculate treewidth using $treewidth - min - degree$ function from NetworkX [27], which calculates using the Minimum Degree heuristic.

- **Centrality** specifies how important a node is within a graph. Following [22], Betweeness Centrality $BE = \sum_{jk} \frac{p_{x_j}, x_k(x_i)}{p_{x_j}, x_k}$ is studied, where $p_{x_j}, x_k(x_i)$ is the number of paths that pass through $x_i$ ,and $p_{x_j}, x_k$ is the total number of shortest paths from node $x_j$ to $x_k$.

- **Community structure** [5] is measured by modularity($Q$). The modularity $Q$ of $G$ is a value from 0 to 1, computed with respect to a given partition $C$ of the same graph and is a measure of the fraction of within-community edges in relation to *another* random graph that has an equal number of vertices and degree. A graph's modularity score corresponds to the maximal modularity of any possible partition in C: $Q(G) = \max\{Q(G, C) | C\}$. While computing the exact value of $Q$ is NP-hard, it is possible to approximate lower-bounds to $Q$, which can be calculated as

$$Q(G, C) = \sum_{C_i \in C} \frac{\sum_{x,y \in C_i} w(x,y)}{\sum_{x,y \in V} w(x,y)} - \left( \frac{\sum_{x \in C_i} \deg(x)}{\sum_{x \in V} \deg(x)} \right)^2, \tag{1}$$

where $w(x, y)$ is the weight between vertex $x$ and $y$.

- **Small-world** [58] measures the extent of graph topology. It is approximated using Proximity Ratio ($P_r$). A graph is considered small-world if $P_r \geq 1$. In this work we use the sigma function from NetworkX [27] for calculation, which is only an approximation of the $P_r$ value due to the need for random graph generation.

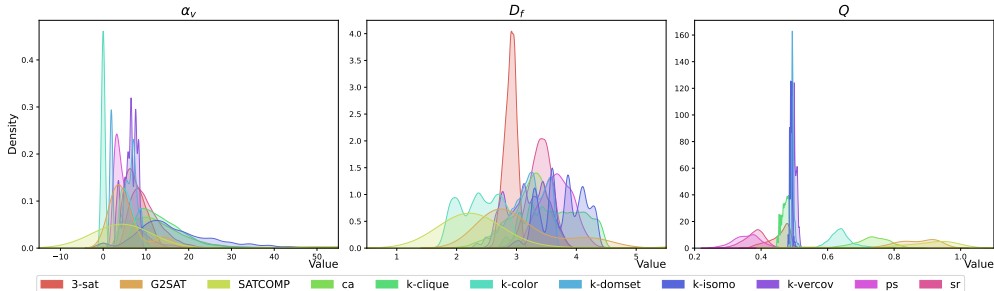

Figure 2: Distributions of various structural graph properties from different problem domains. Left: scale-free measure by variable $\alpha_v$, Middle: fractal dimension $D_f$, Right: modularity $Q$.

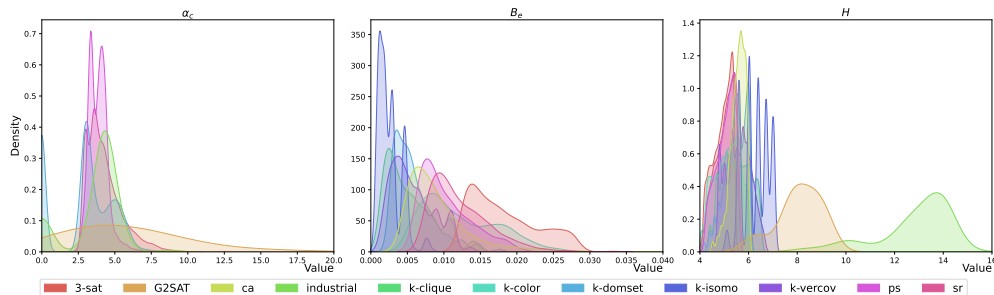

Figure 3: Distribution of various structural graph properties from different problem domains. Left: scale-free measure by clause $\alpha_c$. Middle: centrality measure $BE$. Right: entropy measure $H$.

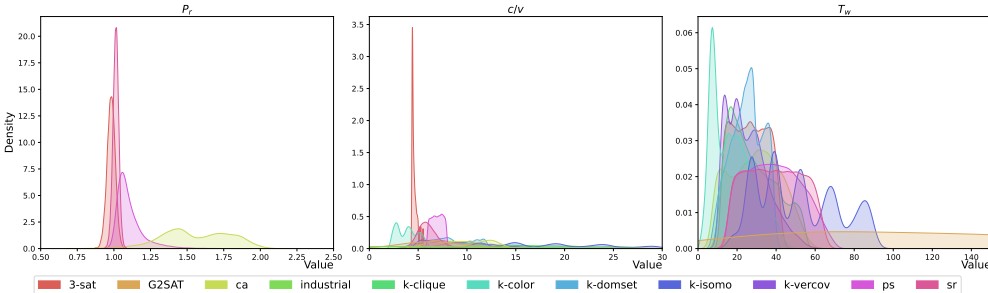

Figure 4: Distribution of various stuctural graph properties from different problem domains. Left: Proximity:$P_r$. Middle: ratio $c/v$. Right: Treediwth $T_w$.

- **Entropy** [66] measures the uncertainty of random systems. Given volume of the graph $vol_G$, we measure the One-dimensional Entropy ($H$) following [66], which can be defined as

$$H = - \sum_{i=1, x_i \in X}^{|X|} \frac{\deg(x_i)}{vol_G} \log \frac{\deg(x_i)}{vol_G}. \tag{2}$$

## B.2  StructureSAT composition

### B.2.1  Dataset generation

We use various existing SAT generators and datasets to produce original SAT formulas in 11 domains. To analyse the property distributions in different domains, we generate 10k problem pairs for selected domains, with each pair containing 50% satisfiable and 50% unsatisfiable problems. Although the generated problems are easy for traditional SAT solvers to solve, they contain a variety of properties and are efficient for modern GNNs to learn on, which provide enough insight for our purpose. Fig. 2 shows the distributions of various structural properties described in Appendix B.1 from calculating the statistical values on raw data pairs with a 60 seconds timeout. For industrial SAT problems and

graph generative model generators, less than 100 pairs are selected. In the next section we detail the provided domains in StructureSAT, including important parameters used for generation.

**Random-3-SAT**: Uniform random 3-SAT ($r$-3SAT) is a special case for Random $k$-SAT where each clause contains exactly 3 literals. The problems are generated using CNFgen [40] with $10-40$ number of variables at the phase transition point.

**SR($n$)**: SR($n$) [53] is a special expression of random $k$-SAT that consists of a balanced dataset with pairs, with $k$ as the maximum number of variables in the problem. Each pair contains one satisfiable problem and one unsatisfiable problem, differing by 1 single literal in one clause. In this work we use problem with $10-40$ number of variables as main training data, represented as SR(10-40).

**Combinatorial problems** (5 domains): For most combinatorial problems, the goal is to find some combination of elements in a solution space while respecting defined constraints. A valid solution would correspond to a satisfying assignment in the SAT encoding of the problem [9]. In StructureSAT, we generate 5 combinatorial problems using CNFgen [40]. Each problem is related to solving constraints over a graph. Thus, we generate random graphs using the Erdős–Rényi model [48] with edge probability $\binom{v}{k}^{-1/\binom{v}{2}}$, where $v$ representing number of vertex. The selected parameters and problems include $k$-coloring, $3 \leq k \leq 4$, $k$-dominating-set, $2 \leq k \leq 3$, $k$-clique-detection, $3 \leq k \leq 4$, $k$-vertex-cover, $3 \leq k \leq 5$, and automorphism in graph $G$.

**CA**: Community Attachment (CA) is a seminal work generated from [24] to mimic the community structure of industrial problems, i.e., value of *modularity $Q$* on a SAT instance's VIG graph. In StuctureSAT, we select 0.7-0.9 as the value of $Q$. The generator uniformly and randomly selects 4 to 5 literals inside the same community with probability $P = Q + 1/co$, where $co$ is the number of communities, and 4 to 5 literals in distinct community with probability $1 - P$. These process are done iteratively to form a formula.

**PS**: Popularity-Similarity (PS) [25] is a problem generation algorithm developed on the idea of *locality*, which is a measure of both community structure and scale-free structure on a SAT instance's VCG graph. PS randomly samples variable $v$ in clause $c$ with the probability $P = 1/(1 + n_v^\beta * n_c^{\beta_1} * \theta_{n_v * n_c}/R)^T$, where $\beta$ is the power-law distribution, $\theta$ is random angle assigned to $v$ and $c$, $T$ is the temperature between 0.75 and 1.5, and $R$ is an approximate normalisation constant.

**SATCOMP**: StructureSAT uses industrial instances from SATCOMP 2007.

**G2SAT**: To overcome the limiting number of instances in industrial problem, graph generative models have been used as problem generator. In this work we mainly use **G2SAT** [63] as generator. Following previous work, we select 20 problems from the industrial dataset, standadise with SatElite preprocessor[20], and generate similar instances using GraphSAGE [28] and a two-phase generation process. Both industrial and G2SAT datasets are used mainly as testing set in the dataset.

### B.3 Augmented problems

This section describes augmented dataset setup that might affect the generalisation ability of GNN. CDCL-based SAT solvers produce conflict clauses during searching and add learned clauses, which are reverse of conflict clauses, to the original problems. Prior work [44] has shown that training on augmented SAT problems with the learned clauses added could lead to better accuracy on augmented SAT domains than training on raw problems. Since this might be caused by the destroying of structure during the addition of learned clauses [5], we are interested in an empirical analysis on the structural difference and their effect on out-of-distribution generalisation in this work. Thus, we gather learned clauses from produced DRAT-trim proofs [60] after running problems with the CadiCal solver [21], then augment our dataset with the collected learned clauses.

### B.4 Structure-aware splitting and generalisation selection

We also propose a novel splitting method for StructureSAT and focus on 3 types of generalisation tasks. After generating the raw datasets, instead of randomly splitting them to train/valid/test sets, we divide each raw datasets based on specific structural values. Specifically, for each domain, we select the average value $Z$ from every calculated graph-based properties, as shown in Table 3, and

Table 3: Mean structural properties of StructureSAT domains. N/A indicates unavailable.

| | $D_f$ | $\alpha_v$ | $\alpha_c$ | $T_w$ | $B_e$ | $Q$ | $H$ |
|---|---|---|---|---|---|---|---|
| SR (10-40) | 3.39 | 4.25 | 10.03 | 39.12 | 0.012 | 0.38 | 5.01 |
| r-3SAT | 2.90 | 7.54 | N/A | 26.53 | 0.018 | 0.45 | 4.92 |
| $k$-clique | 3.67 | 13.44 | N/A | 28.19 | 0.005 | 0.47 | 5.51 |
| $k$-dominating-set | 3.19 | 5.44 | 2.95 | 25.03 | 0.006 | 0.49 | 5.46 |
| $k$-vertex-cover | 3.18 | 6.29 | N/A | 25.24 | 0.006 | 0.50 | 5.58 |
| $k$-coloring | 2.38 | 2.92 | N/A | 17.73 | 0.02 | 0.64 | 5.13 |
| automorph | 3.78 | 17.47 | N/A | 51.25 | 0.0025 | 0.49 | 6.28 |
| CA | 3.19 | 12.13 | N/A | 29.39 | 0.009 | 0.73 | 5.56 |
| PS | 3.68 | 5.49 | 3.87 | 38.66 | 0.01 | 0.36 | 5.06 |
| G2SAT | 3.06 | 5.30 | 5.56 | 116.87 | N/A | 0.87 | 8.09 |
| industrial | 2.68 | 10.09 | 3.50 | N/A | N/A | 0.87 | 12.49 |

split each domain into a low-value subset and a high-value subset, producing a total of 16 subsets (8 structures $*$ 2 ranges) per domain. Each subsets contains 80k training data and 10k validation data.[2]

To test the generalisation GNNs, we focus on 3 aspects: in-domain larger problems, in-domain problems with different properties, and out-of-distribution problems. For all the synthetic generators, we follow the generation process described in Appendix B.2.1 and randomly generate problems with 10-40, 100-200, and 200-300 number of variables, each with 10k pairs. To test generalisation on uniform size problems with different structures, we are interested in random 3-SAT with different $c/v$ ratios and different backdoor/graph properties evaluation on all domains. When calculating phase transition on random 3-SAT with 10-40 number of variables, the ratio value is roughly 4.7. To make the generation process easier, our generated test set has a $c/v$ ratio of 3.5, 4, 5.5 and 6, as the further away from phase transition point, the harder it is to generate balanced dataset. For out-of-distribution problems, we randomly generate 10k problem testing data from each synthetic generators without property split, thus holding the structural range as in Fig. 2, Fig. 3 and Fig. 4. Small-size industrial and G2SAT problems are also selected as part of testing set.

## C   Full analysis on Dataset

To get a comprehensive understanding of base dataset domains without splitting, we extend Fig. 2 and plot 6 other graph property distributions in Fig. 3 and Fig. 4 including $\alpha_c$, $H$, $BE$, $P_r$, $T_w$ and $c/v$ ratio. Since LCG is a bipartite graph, some calculations have been adjusted to accommodate its bipartite characteristics. For example, the clustering coefficient in $P_r$ has been modified to use the bipartite clustering coefficient calculation.

The figures show a wide variety of distributions over graph properties among different domains. Some distinguishable differences can be seen within $Q$ in Fig. 2, where random, crafted, and industrial problems have low, medium, and high values, respectively. Additionally, industrial and crafted domains have a wider range of $D_f$ values than random domains. In Fig. 3, the value range of $B_e$ ranks from lowest to highest in crafted, industrial, and random domains, while industrial problems have highest $H$ values. For available $P_r$ values in Fig. 4, the values from random domains 3-SAT and SR are close to 1, while the pseudo-industrial problems CA and PS have higher $P_r$. All the distinct characteristics within the dataset highlight potential of affecting generalisation differently with GNNs.

## D   Limitations

Despite being the first dataset to more deeply analyze SAT structure in terms of GNN generalisation, there are still a few limitations and many possibilities to consider for future work. Firstly,

---

[2]Our dataset and codebase are available here.

Table 4: NeuroSAT testing results. For each metric, we train NeuroSAT on the small and big split of the corresponding SR(10-40) training sets. We then test the trained model on the testing data of 10-40, 40-100, and 100-200 variables for each metric. Better training split performance in bold.

| Train Split on SR(10-40) | Test | Metric | | | | | |
|---|---|---|---|---|---|---|---|
| | | $D_f$ | $\alpha_v$ | $\alpha_c$ | $T_w$ | $Q$ | $H$ |
| small | 10-40 | 94.49 | **94.40** | **95.41** | 95.04 | 93.76 | 94.00 |
| big | | **95.43** | 93.05 | 95.31 | **95.33** | **94.81** | **95.77** |
| small | 40-100 | 68.51 | **69.68** | 70.78 | 65.45 | 70.83 | 64.82 |
| big | | **72.41** | 67.09 | **73.22** | **72.14** | **73.32** | **73.78** |
| small | 100-200 | **57.04** | **56.78** | 57.74 | 54.72 | **57.93** | 53.77 |
| big | | 56.98 | 54.5 | **58.98** | **56.47** | 57.78 | **57.98** |

StructureSAT is separated into subsets, where each subset has one domain and focuses on one type of feature only. While useful for identifying individual factors of influence at a property level, a multivariate analysis of features and domains could be considered in future work. Secondly, due to certain problems being fundamentally intractable, we do not consider CNF based features like backdoor[37]. The structure calculations in Fig. 2 are limited to a 60 seconds timeout as most industrial and G2SAT generative processes are computationally hard to compute. Exploring effective algorithms of computing the structural properties on SAT, especially large problems, could be a potential future direction. Thirdly, although we include SATCOMP2007 in our dataset and calculate structural values of individual problems, the models we trained could not be efficiently tested on large industrial and G2SAT problems. Thus we only test on 24 G2SAT problems and 12 small industrial problems used to generate the G2SAT problems inside experiment. However, either better models or larger training sets could be included in future work for GNNs to solve large industrial problems. Lastly, as we only generated 100k pairs of data for each domain and divided inside the base dataset, there is overlap between different property split groups. Potential future work could be generating datasets with more controllable feature values as what CA generator do [24].

# E  More experiment details

## E.1  GNN baseline

Models used in this work include NeuroSAT [53], GCN [38] and GIN [61].

## E.2  Code Base, experiment set up, and license

Our Dataset and Code base are available in StructureSAT[1]. The link also include detailed instruction on downloading and running experiment with the dataset.

Experiments with GNN are done using existing work from [44]. Experimental parameters include $1e-04$ learning rate, $1e-08$ weight decay and 32 number of message passing iterations. All the experiments are run on a machine with a NVIDIA A4500.

StructureSAT is openly licensed via CC BY 4.0.

# F  Additional Experiment Results

This section details some additional experiments. In this section, for simplicity, we make SR stand for SR(10-40), R3 stand for Random-3-SAT, KCL stand for $k$-clique, KD stand for $k$-dominating-set, KV stand for $k$-vertex-cover, KCO stand for $k$-coloring, AM stand for automorphism, G2 stand for G2SAT, and IN stand for industrial problems. Note that for simple computation, we select 24 small problems from G2SAT. 12 industrial problems used to train G2SAT are used as industrial problems in this section.

---

[1]https://drive.google.com/drive/folders/1ZrhRlRqQUTrYExVzVbXaocjvNIi2Os25?usp=sharing.

### F.1 Generalisation to in-domain problems

**Large size generalisation.** To answer Q1, we train a NeuroSAT on a small SAT problem set and test its performance on varying number of variables. Table 4 shows the testing result of NeuroSAT trained on SR(10-40), which is SR(n) problems with 10-40 number of variables. In general, NeuroSAT struggles to generalise to larger problems with more variables (e.g. 40-100 and 100-200). Although there is no major difference among most structure measures, models trained on larger property values (e.g. the big (10-40) splits) generalise better to large problems in the same domain compared to the small splits. We hypothesize this may due to the increase in relevant metric values in larger problems. Thus, generating problems with property values in the upper range, especially larger $H$ and $\alpha_c$ values as seen in Table 4 has the potential to enable GNNs solve larger in-domain problems.

Table 5: r-3SAT testing results with different ratios.

| Metric | Accuracy |
|--------|----------|
| 3.5 | 97.6 |
| 4 | 95.87 |
| 4.7 | 95.5 |
| 5.5 | 98.7 |
| 6 | 99.37 |

**Different ratio generalisation.** Table 5 shows the result of NeuroSAT trained on random 3-SAT and tested on a varying $c/v$ ratio, through which we try to answer Q2. Surprisingly, although models are trained at the phase transition, which is the "hardest" point for random 3-SAT as discussed in Appendix B.1, testing at the phase transition point has the lowest accuracy. Furthermore, the further away $c/v$ is from the phase transition point, the better their testing accuracy. This signals that GNNs are not always better at solving problems with similar properties to the training data.

**Out-of-domain generalisation** Below are extended results for out-of-domain generalisation.

Table 6 shows the result of GCN model trained on SR(10-40), splitted by 3 graph properties, while Table 8 shows the result of GIN model with same training set.

Table 7 shows the result of NeuroSAT model trained on random 3-SAT problems splitted by 5 graph properties.

Table 9 shows the result of NeuroSAT model trained on SR problems while predicting satisfying assignment.

Table 6: Accuracy of GCN trained on SR(10-40) with different structural splits.

| Metric | Split | Testing Domains | | | | | | | | | | |
|--------|-------|-------|-------|-------|-------|-------|-------|-------|-------|-------|-------|-------|
| | | SR | R3 | KCL | KD | KV | KCO | AM | CA | PS | G2 | IN |
| $\alpha_v$ | small | 93.20 | 86.42 | 51.16 | 53.96 | 58.42 | 45.51 | 52.95 | 60.84 | 94.67 | 79.17 | 58.33 |
| | big | 92.25 | 84.72 | 50.10 | 49.91 | 50.01 | 50.00 | 50.00 | 55.69 | 94.69 | 87.50 | 50.00 |
| $\alpha_c$ | small | 95.41 | 92.99 | 51.85 | 59.27 | 68.10 | 55.97 | 43.85 | 92.36 | 96.04 | 87.50 | 58.33 |
| | big | 95.31 | 92.82 | 48.83 | 56.14 | 54.97 | 51.75 | 45.79 | 93.31 | 96.06 | 91.67 | 75.00 |
| $Q$ | small | 85.39 | 71.8 | 50.93 | 56.69 | 58.88 | 49.89 | 47.61 | 52.16 | 89.32 | 95.80 | 66.67 |
| | big | 88.12 | 72.65 | 49.55 | 48.71 | 48.89 | 49.62 | 50.64 | 54.83 | 91.01 | 95.80 | 50.00 |

**Comparison with Traditional Solver** In the following section we compare the performance of traditional solver CDCL with one of the NeuroSAT models trained on SR(10-40) splitted on small $\alpha_c$, and show the results in Table 10. The NeuroSAT model was tested with a batch size of 212 on the AM domain, 412 on KCL, and 512 on all other domains. Both the CDCL and NeuroSAT solvers were tested on the same machine to ensure consistency in comparisons. Regarding solving accuracy, for the tested datasets generated by random generators, the traditional CDCL solver achieved 100% accuracy, while NeuroSAT, trained on SR(n), reached approximately 50% accuracy on certain domains. In terms of solving time, NeuroSAT, with appropriately selected batch sizes, demonstrated the potential to solve problems faster than CDCL solvers on both random and crafted instances. Additionally, CDCL's solving time are generally similar, with variations occurring depending on the domain type (random vs. crafted), while the GNN solver's performance was more sensitive to the domains themselves. For instance, the CDCL solver exhibited similar solving times across random problems SR and R3, while solving times across combinatorial domains (KCL, KD, KV) were consistent yet distinct from random domains. In contrast, NeuroSAT solved R3 significantly faster than SR, despite both being categorized as random domains, which is likely due to differences in the size and complexity of the input graphs across domains. This also suggest the complete different solving

Table 7: Accuracy of NeuroSAT trained on R-3SAT with different structural splits.

| Metric | Split | Testing Domains | | | | | | | | | | |
|--------|-------|-------|-------|-------|-------|-------|-------|-------|-------|-------|-------|-------|
| | | SR | R3 | KCL | KD | KV | KCO | AM | CA | PS | G2 | IN |
| $D_f$ | small | 58.55 | 95.02 | 51.78 | 49.69 | 48.76 | 50.00 | 47.70 | 69.06 | 69.89 | 37.5 | 75.00 |
| | big | 56.77 | 96.19 | 50.00 | 48.19 | 50.00 | 40.42 | 50.00 | 69.93 | 73.93 | 20.83 | 66.67 |
| $\alpha_v$ | small | 59.80 | 94.84 | 50.00 | 50.00 | 50.00 | 42.22 | 50.00 | 47.32 | 77.92 | 91.67 | 75.00 |
| | big | 58.07 | 95.48 | 50.00 | 49.86 | 50.00 | 40.83 | 50.00 | 62.79 | 73.54 | 20.83 | 75.00 |
| $T_w$ | small | 57.58 | 93.77 | 51.70 | 49.82 | 50.00 | 43.89 | 46.97 | 50.00 | 63.62 | 37.50 | 50.00 |
| | big | 55.23 | 95.29 | 50.00 | 48.99 | 50.00 | 41.41 | 46.35 | 57.33 | 58.34 | 66.67 | 75.00 |
| $Q$ | small | 56.68 | 92.12 | 50.35 | 50.00 | 50.00 | 50.00 | 64.15 | 48.16 | 72.15 | 29.17 | 50.00 |
| | big | 55.93 | 95.80 | 50.05 | 48.99 | 50.00 | 44.50 | 59.79 | 48.69 | 57.29 | 16.67 | 75.00 |
| $H$ | small | 57.33 | 94.26 | 48.83 | 49.21 | 50.00 | 40.97 | 54.79 | 50.00 | 71.67 | 50.00 | 41.67 |
| | big | 57.28 | 95.24 | 50.00 | 49.99 | 50.00 | 47.75 | 50.00 | 78.16 | 69.74 | 33.33 | 75.00 |

Table 8: Average accuracy of GIN trained on SR(10-40) with different structural splits.

| Metric | Split | Testing Domains | | | | | | | | | |
|--------|-------|------|------|------|------|------|------|------|------|------|------|
| | | R3 | KCL | KD | KV | KCO | AM | CA | PS | G2 | IN |
| $D_f$ | small | 92.0 | 53.3 | 64.6 | 65.0 | 49.0 | 50.4 | 75.7 | 91.5 | 47.2 | 86.1 |
| | big | 88.3 | 57.2 | 62.5 | 59.2 | 52.2 | 46.8 | 64.6 | 88.1 | 61.1 | 63.9 |
| $\alpha_v$ | small | 93.5 | 51.6 | 55.3 | 55.9 | 47.6 | 47.4 | 65.5 | 93.8 | 50.0 | 80.6 |
| | big | 92.3 | 53.4 | 59.0 | 65.5 | 50.0 | 52.3 | 70.5 | 93.7 | 50.0 | 63.9 |
| $\alpha_c$ | small | 93.3 | 54.3 | 60.1 | 65.5 | 59.6 | 50.3 | 68.0 | 95.4 | 58.3 | 79.2 |
| | big | 93.4 | 51.7 | 54.8 | 59.0 | 56.7 | 49.0 | 65.0 | 92.1 | 52.8 | 57.0 |
| $Q$ | small | 90.8 | 48.8 | 59.9 | 60.8 | 49.4 | 53.6 | 63.6 | 91.6 | 50.0 | 86.1 |
| | big | 89.9 | 53.2 | 63.2 | 62.5 | 50.0 | 52.7 | 65.6 | 86.9 | 38.7 | 68.1 |

Table 9: Accuracy of NeuroSAT trained on SR(10-40) with different structural splits, task is satisfying assignment prediction.

| Metric | Split | Testing Domains | | | | | | | | | | |
|--------|-------|-------|-------|-------|-------|-------|-------|-------|-------|-------|-------|-------|
| | | SR | R3 | KCL | KD | KV | KCO | AM | CA | PS | G2 | IN |
| $\alpha_v$ | small | 94.40 | 91.72 | 48.76 | 66.03 | 74.89 | 44.70 | 46.98 | 90.28 | 94.44 | 95.83 | 50.00 |
| | big | 93.05 | 91.40 | 51.60 | 53.21 | 53.64 | 56.14 | 58.37 | 70.51 | 91.11 | 83.33 | 50.00 |
| $\alpha_c$ | small | 95.41 | 92.99 | 51.85 | 59.27 | 68.10 | 55.97 | 43.85 | 92.36 | 96.04 | 87.50 | 58.33 |
| | big | 95.31 | 92.82 | 48.83 | 56.14 | 54.97 | 51.75 | 45.79 | 93.31 | 96.06 | 91.67 | 75.00 |
| $Q$ | small | 93.76 | 91.87 | 56.30 | 71.76 | 51.9 | 55.89 | 49.75 | 89.08 | 95.75 | 19/24 | 50.00 |
| | big | 94.81 | 92.74 | 51.61 | 61.06 | 60.80 | 61.64 | 46.57 | 89.99 | 93.00 | 83.33 | 58.33 |

Table 10: Solving time (in seconds) comparison between traditional solver and NeuroSAT.

| Solver | Metric | Testing Domains | | | | | | | | |
|--------|--------|-------|-------|-------|-------|-------|-------|-------|-------|-------|
| | | SR | R3 | KCL | KD | KV | KCO | AM | CA | PS |
| CDCL | Time | 57.92 | 56.90 | 61.87 | 61.58 | 63.31 | 58.25 | 64.40 | 61.78 | 59.92 |
| GNN | Time | 45.56 | 16.52 | 44.92 | 31.00 | 40.54 | 21.07 | 93.80 | 33.24 | 22.35 |

heuristics between CDCL and GNN models, and for GNN model performance, analysing on each individual domains, instead of a general type of domain, is necessary.

