# OpenReview forum: "Structure Based Dataset on SAT Solving with Graph Neural Networks"
_NeurIPS.cc/2024/Workshop/MATH-AI — MATH-AI 24_

### Official Review · Reviewer_ktDc · 2024-09-27
**A Good Benchmark for Evaluating the Impact of Structural Properties on GNN-based SAT Solvers**

**Rating:** 6
**Confidence:** 5

**Review:**

This paper introduces StructureSAT, a large benchmark designed to evaluate the generalization capabilities of GNN-based SAT solvers across various structural properties of SAT instances. The comprehensive experiments demonstrate that different structural properties can influence the generalization ability of GNN-based SAT solvers to some extent.

While I find the exploration of structural properties in SAT instances intriguing and commend the efforts in dataset construction and evaluation, the paper's focus is primarily on empirical performance evaluation. However, it appears challenging to derive a unified conclusion about how GNN-based SAT solvers perform consistently across instances with large or small values of these properties. Additionally, it remains unclear how these insights can be leveraged to enhance or optimize the performance of GNN-based solvers when encountering specific ranges of these structural properties.

---

### Official Review · Reviewer_iSFP · 2024-10-05
**Structure Based Dataset on SAT Solving with Graph Neural Networks**

**Rating:** 6
**Confidence:** 3

**Review:**

**Summary**: This paper introduces StructureSAT, a dataset of SAT problems, along with code to generate SAT problems. It also presents a method for splitting the dataset into more detailed hierarchies based on structural properties. The authors conduct various experiments to evaluate the generalization capabilities of different GNN solvers using this dataset. The results showcase existing challenges and suggest potential future directions for enhancing generalization performance.

**Pros**: The paper provides a novel perspective on improving GNN-based SAT solvers. The dataset is well-designed and covers various domains, and the authors conduct a sufficient number of experiments. Additionally, the writing is clear and well-structured.

**Cons**: The paper may be vague for those outside the field and could benefit from providing more detailed explanations.

---

### Official Review · Reviewer_2r5D · 2024-10-05
**This paper introduces StructureSAT, a novel dataset for SAT solving with Graph Neural Networks (GNNs) that focuses on the relationship between structural properties and GNN generalization. The authors propose a structure-aware splitting method and conduct experiments to evaluate GNN performance across various problem domains and structural properties, aiming to bridge the gap between traditional SAT solving techniques and modern machine learning approaches.**

**Rating:** 7
**Confidence:** 3

**Review:**

This paper presents a novel and important contribution to the field of SAT solving with machine learning techniques. The introduction of StructureSAT and the insights gained from the generalization experiments are valuable to the research community. While there are areas for improvement, such as more detailed comparisons with traditional solvers and deeper error analysis, the overall quality and significance of the work merit acceptance. The paper opens up several interesting avenues for future research and has the potential to influence the development of more effective GNN-based SAT solvers.

### Quality:
##### Strengths:
- The paper addresses an important and timely topic in the intersection of SAT solving and machine learning.
- The methodology is sound and well-justified, incorporating both CNF-based and graph-based structural properties.
- The experimental design is comprehensive, covering various aspects of generalization.

##### Weaknesses:
- The paper lacks a detailed comparison with state-of-the-art CDCL solvers, which could provide more context for the GNN performance.

### Clarity of writing:
##### Strengths:
- The paper is generally well-structured and easy to follow.
- The authors clearly explain their motivation and approach.

##### Weaknesses:
- Some technical details, particularly in the description of structural properties, could be explained more clearly for readers less familiar with SAT solving terminology.

### Originality of the paper:
##### Strengths:
- The focus on structural properties and their impact on GNN generalization is novel and potentially impactful.
- The structure-aware splitting method is an innovative approach to dataset creation.

##### Weaknesses:
- While the combination of ideas is novel, many of the individual components (GNNs for SAT, structural analysis of SAT problems) have been explored separately in previous work.

### Significance of this work:
##### Strengths:
- The introduction of StructureSAT as a large-scale, diverse dataset is a significant contribution to the field)
- The insights gained from the generalization experiments could influence future development of GNN-based SAT solvers.

##### Weaknesses:
- The paper could elaborate more on the practical implications of the findings for real-world SAT solving applications.

---

### Official Review · Reviewer_ibip · 2024-10-07

**Rating:** 7
**Confidence:** 4

**Review:**

The paper presents StructureSAT, which is a novel dataset that consists of SAT problems of various sources and types. One highlight of the paper is the investigation on the generalization capability of GNN solvers. However, the paper could benefit from more theoretical analysis and more extensive experiments (particularly, more GNN baselines and more state-of-the-art GNN variations that have better expressivity or generalization performance).

---

### Decision · Program_Chairs · 2024-10-09

**Decision:**

Accept

**Comment:**

All reviewers agree that this work provides an interesting benchmark for SAT solving with GNNs, and the topic would likely be of interest to the MATH-AI community.